

# Analysis of the anthropogenic effect on *the Silencio* River in Salvador Escalante, Michoacán, México

Mario Alberto Pérez-Méndez[1], Guadalupe Selene Fraga-Cruz[1], Gustavo Álvarez-Gómez[1], Fabricio Nápoles-Rivera[1], Gladys Jiménez-García[2] and Rafael Maya-Yescas[1]

[1] Facultad de Ingeniería Química, Universidad Michoacana de San Nicolás de Hidalgo, Morelia, Michoacán, Mexico
[2] Academia de Ingeniería Biomédica, Instituto tecnológico superior de Pátzcuaro, Pátzcuaro, Michoacán, Mexico

## ABSTRACT

The average annual water availability worldwide is approximately 1,386 trillion cubic hectometers ($hm^3$), of which 97.5% is saltwater and only 2.5% is freshwater. Nearly 70% is not available for human consumption as it is in glaciers, snow, and ice. It is estimated that only 0.77% is accessible freshwater for human use. Mexico has an availability of 451,584.7 million cubic meters ($m^3$) of freshwater, with accessibility and distribution being unequal. The growth in urbanization, population, and industrialization has caused a decrease in water quality, and other parameters. Organic and inorganic contaminants evolved from various sources cause the degradation of water quality. The pollution of aquatic bodies, such as rivers and lakes, is one of the main problems in the world. In Salvador Escalante in México, the domestic wastewaters treatment plant (WWTP) is being exposed to effluents contaminated with metals like copper, cadmium, lead, and mercury. In this work, active sludges from the WWTP were analyzed. First, particle size distribution of flocs was measured by a sedimentation process. Secondly, analysis of the tolerance that microorganisms exhibit to metals (*i.e.*, factors) was performed, based on a $2^{(4-1)}$ factorial design of experiments at laboratory-scale, measuring pH, chemical oxygen demand (COD) and electrical conductivity (responses). This aims to evaluate the capacity of the WWTP for improve the water quality. Microbiologic cultures were used for a qualitative study of the microorganisms contained in the active sludges; it was found that *Enterobacterium* does not grow in presence of heavy metals. Cadmium is the most harmful metal for microorganisms according to Pareto diagrams presented in this study.

## INTRODUCTION

The impact of human activities on water resources has increased due to the worldwide population growth and the subsequent expansion of urban, industrial, and agricultural sectors in both developed and developing countries which has important consequences for the ecosystem and human health (*Vázquez-Tapia et al., 2022*; *Chakraborti & Shimshack,*

Corresponding author
Mario Alberto Pérez-Méndez,
mario.perez@umich.mx

*2022*). A multitude of factors have consequently increased the concentration of heavy metals and organic pollutants in the water, leading to deterioration of water quality, and making future water supply more uncertain (*Fabian, 2012*; *Li, Li & Li, 2021*; *Zhang et al., 2020*).

In the aerobic process, the microbes would reduce the organic matter in presence of oxygen in water. The organic matter load is rapidly utilized by microbes, a mechanism that has led to the development of biological wastewater treatment methods globally (*Varila-Quiroga & Díaz-López, 2008*). However, in several countries, wastewaters treatment plants (WWTPs) are not designed efficiently, leading to poor water purification within them, resulting in poor quality effluent being dumped into lakes and rivers (*Macedo et al., 2022*). Many parts of the world face severe water shortages, wastewater reuse methods. Some researchers show that about 730 megatons of waste are annually discharged into water from sewages and other effluents (*Priya et al., 2021*).

In addition to this, there are emerging contaminants in water such as pharmaceuticals and microplastics. Because, the low degradation capacity and the complications for the reuse of bottles and plastic bags, every year there is more plastic in the water bodies (*UNEP, 2016*). However, all plastics are exposed to the microplastic and nanoplastic concentration in water and have been recognized as an emergent environmental problem (*Pastor & Agullo, 2019*). Microplastics are tiny particles of plastic material, typically measuring less than 5 millimetres in size; most of them are formed due the breakdown of larger plastic items degraded by many factors like sunlight, wind and water. This process will continue with the time generating, smaller and smaller particles. Also, the microplastic interaction with the trace elements will have adversal effects in the environment (*Cao et al., 2021*; *Verla et al., 2019*).

In Mexico, freshwater bodies are exploited for consumption and storage, including dams, rivers, and lakes. Specifically, Lake Zirahuen is currently affected by heavy metals like mercury (Hg), lead (Pb), copper (Cu), and cadmium (Cd) and by the microplastics coming from Santa Clara del Cobre by the river El Silencio (Fig. 1). One of the main activities in the municipality is copper craftsmanship. The arrivals of recent technologies have increased production of copper workshops and, consequently, increased the amount of wastewater, which is discharged to the river El Silencio. The presence of metals including Hg, Pb, Cd, and others, even in trace amounts, may cause health problems (*Jaishankar et al., 2014*; *Boskabady et al., 2018*).

The presence of lead in plants can cause damage to ther lipid membrane, damaging to chlorophyll and photosynthetic processes, thus altering the growth of plants in general (*Najeeb et al., 2014*). The ionic mechanism of lead toxicity occurs mainly due to the ability of lead metal ions to replace either bivalent or monovalent cations, this can cause some disturbs the cell metabolism (*Jaishankar et al., 2014*). Some of the reported problems with the lead are biological processes such as cell adhesion, intra- and inter-cellular signaling, protein folding, maturation, apoptosis, *etc.* (*Johnson et al., 2017*; *Lang et al., 2020*). Lead bioaccumulation is dangerous for human health due to induction of inflammatory cascades in tissues and organs. The lead provokes mainly respiratory, neurologic, digestive,

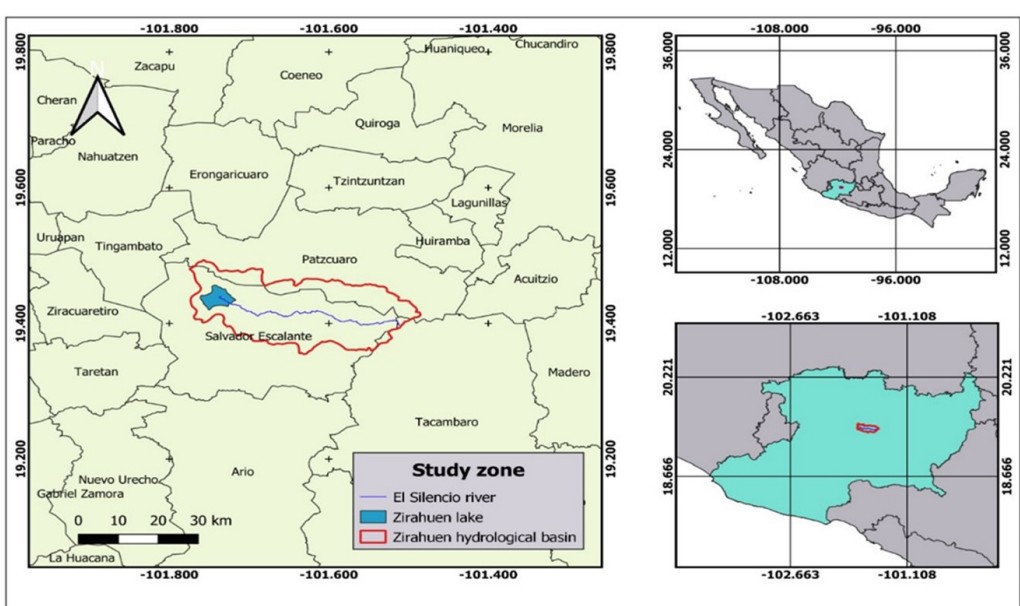

**Figure 1** **Location of area under study.** Map base layer from https://ceac.michoacan.gob.mx/wp-content/uploads/2016/08/Lago-Zirahuen.pdf.

and cardiovascular disorders (*Boskabady et al., 2018*). Data were collected as previously described in *Cabral et al. (2015)*.

Mercury is very toxic and exceedingly bioaccumulative. Major sources of mercury pollution include anthropogenic activities such as agriculture, municipal and industrial wastewater discharges, mining and incineration (*Chen, Chen & Dng, 2012*). Mercury appears in three main forms: metallic elements, inorganic salts and organic compounds. The mercury is absorbed by microorganisms and transformed into methylmercury within them, eventually undergoing biomagnification. The main route of human exposure to methylmercury is by the consumption of aquatic animals (*Trasande, Landrigan & Schechter, 2005*). Due to their long biological half-life and their cumulative properties, long-term exposure to Cd and Pb may cause chronic adverse health effects. Metal toxicity can be direct as well as indirect by inducing the production of reactive oxygen species (ROS) that can cause damage to cell macromolecules, and notably polyunsaturated lipids and proteins (*Cabral et al., 2015*). On the other hand, cadmium has the ability to bind with cystein, glutamate, histidine and aspartate ligands and can lead to the deficiency of iron (*Castagnetto et al., 2002*; *Wang et al., 2019*). Also, the adsorption of positively charged ions is inversely related to salinity of water: as salinity increases, metal species ions compete for the adsorption onto plastics. This has been observed for Pb, Cd, Zn and Cu (*Fu et al., 2020*; *Tang et al., 2020*). Salts of each studied metal will be added at different concentrations according to a factorial experimental design described later.

The flocculation capacity measurement is a good indicator of activated sludge in a WWTP, since a floccus sediment represents bacterial growth and is strictly related to the degradation of the organic matter present in the water. On the other hand, bacterial growth

is affected by the presence of undesirable contaminants; there is less or no increase in sedimentable solids over time, and this increases the area of the settlers required for the wastewater treatment plant.

Therefore, four main objectives in this work were developed for this study on the WWTP of Salvador Escalante municipality: First, following flocculation of sewage sludges, determining operating regions regulated by addition of nutrients or dilution of concentration of mercury, lead, copper, and cadmium. Then, the influence of changes in metals concentration in the aqueous medium on cleanliness and discharge quality of the effluents was considered. Finally, the effect of the presence of those metals in residual water on activity of sludges was measured.

## MATERIALS & METHODS

### Geographical ubication

The WWTP of Salvador Escalante, Michoacán de Ocampo, is located on the road Santa Clara–Zirahuén in the community known as Irícuaro (Fig. 1). The installed capacity of the biodisc or biocontactor filter plant, also known as combined cycle, is 60 L/s and treats 43 L/s (*CONAGUA, 2015*). It is mandatory that the discharge quality from biodisc contact filters meets the standards established in the NOM-001-SEMARNAT-2021 (*Secretariat of Economy of Mexico, 2021*), concerning the maximum allowable limits of contaminants in water discharges to national assets.. The map presented was developed by Álvarez-Gomez using qgis v.3.16.

Due to this situation, it was necessary to install a WWTP to treat the waters of El Silencio river. This plant has been working to improve the treated waters, to reach the requirements according to NOM-001-SEMARNAT-2021 (*Secretariat of Economy of Mexico, 2021*). This document regulates the number of contaminants that can be present in water for agricultural use or irrigation purposes, including factors such as the presence of metals, pH, chemical oxygen demand.

Among the main factors to consider in a WWTP are the presence and health of activated sludge, which degrade the organic matter, improving the water quality. This treated water is discharged, causing as little environmental impact as possible. Furthermore, it is important to analyse the tolerance exhibited by microorganisms present in activated sludge to these metals. Due to the presence of trace elements, it is necessary to study the effect of these heavy metals on microorganisms, and on the quality of the treated water (*Binda et al., 2021*).

As El Silencio River reports Cu concentrations in the range of 0.092–2.87 mg/L in the water whereas the sediment contains about 0.032–1.85 mg/L, it is important to consider that any change in the chemistry of the water can be the trigger for the copper retained in sediments to be released into the water column, thus affecting microorganisms, fish, plants, crops and even riverside localities (*Ayala-Ramirez et al., 2010*). Depending on the season (dry and rain) the contaminants concentration like trace elements (*Mendoza et al., 2015*). It is important to notice the sediment will be reported in ml/L according to NMX-AA-004-SCFI-2013 (*Secretariat of Economy of Mexico, 2013*). A well-mixed sample

was measured in an Imhoff cone up to the 1 L mark and allowed it to settle down for 45 min. The solids adhering to the walls of the cone were then gently detached using a stirrer and left to rest for an additional 15 min, and the volume of settleable solids were recorded in mL/L. If the settleable matter contained pockets of liquid and/or air bubbles among the coarse particles, their volume were approximately estimated and subtracted from the volume of settled solids.

## Microbial tolerance to heavy metal contaminants in the sludge
### Evaluation of oxygen consumption

Oxygen consumption is one of the most important parameters; estimated by cutting the supply thereof and measuring the concentration of oxygen at different times to obtain decay profile, which allows to infer the microbial activity of degradation. The dissolved oxygen was measured with an oxygen-meter (HI9146-04; Hanna Instruments) with a cellophane membrane as a sensor, cutting off the air supply, for the experimental data the use of a fish tank pump used as an oxygen supplier for the system was needed. Measurements were taken every 5 minutes for half an hour to measure the consumption profile by the bacteria itself that would reflect the bacterial activity as well as the time needed to reach anaerobic process. A total of six dissolved oxygen measurements were taken in each bioreactor in the laboratory for each of the experiments proposed in the design, allowing the establishment of the oxygen consumption profile exhibited by the bacteria in the activated sludge.

### Evaluation of chemical oxygen demand

The chemical oxygen demand (COD) was measured using small volumes (two mL) of water samples, which are pipetted into threaded test tubes, containing previously measured reagents (*Secretariat of Economy of Mexico, 2011*). The tubes are incubated until the digestion was complete and then cooled. The COD measurement is performed with a DR 5000 HACH spectrophotometer at 600 nm. Three base measurements were made: at the beginning in the feeding of new raw water to the cell; at 17 h to measure bacterial growth; and at 40 h to confirm if the batch has been changed. The decrease in COD followed the expected behaviour when the sludge is active and functioning. It should be remembered that the COD is an indirect measure of the proper functioning of the bacterial batch responsible for cleaning the wastewater, so that a poor or no decay in the COD over the days will indicate inhibition or death of the microorganism present.

### Evaluation of sedimentable solids

A batch of ''Eckenfelder cells'' were fed every 48 h, with 7.0 L of water from El Silencio river to measure sedimentable solids. These were monitored and increasing their level in the reactor because of degraded organic matter.

The standardized test to determine the sedimentable solids (SS) is to place a sample of residual water in a 1 L Imhoff-type cone and note the volume of solids that settle after a specific period (60 min) (*Secretariat of Economy of Mexico, 2013*). Total settling solids (TSS) represent a grouped parameter. To better understand the nature of the particles that make up TSS in wastewater, it is necessary to measure the size of these particles, which is why the analysis of the particle size distribution is performed. Particle size was determined

**Table 1 Levels of metals in the design of experiments.** The data represents the experiments developed.

| Metal | Minimum, ppm[a] | Central point, ppm | Maximum, ppm |
|---|---|---|---|
| Cu | 3.0 | 4.0 | 5.0 |
| Cd | 0.05 | 0.10 | 0.15 |
| Hg | 0.006 | 0.008 | 0.010 |
| Pb | 1.0 | 2.0 | 3.0 |

**Notes.**

[a] Indexes measured in *Daphnia Magna*, also known as "large sea flea", for 48 h.

with a Beckman Coulter LS100, which measures samples from 0.4 µm to 900 µm. The distribution particle size was then evaluated.

### pH measurement

The measurement and monitoring of pH in a bacteriological system is essential, as many bacteria do not reproduce under certain acidic or alkaline conditions. Therefore, multiple measurements must be taken at 6, 12, 18, and 48 h to evaluate the effect of the presence of metals in the controlled laboratory environment. A Hanna Instrument potentiometer with HI9835 electrode was used for these measurements.

## Design of experiments

A $2^{4-1}$ factorial design of experiments were done (*Montgomery, 2013*) using concentrations of four metals were considered as factors: copper (Cu), mercury (Hg), lead (Pb) and cadmium (Cd). The maximum levels of these were determined based on NOM-001-SEMARNAT-2021 explained before, and their minimum values based on decay index 50 (LD50) (Table 1). As response variables, changes in pH, conductivity, COD and presence of sedimentable solids over time were chosen the maximum level of tolerance to the presence of these metals in the microorganisms was searched. To estimate uncertainty, five replicates on the central point (the media of every criteria measured) were performed. Additionally, qualitative microbial community analyses were performed from samples collected from different points. All the experiments are presented in the Table 2.

## Bacteriological cultures

Bacteriological cultures are a qualitative way to indicate the presence of bacteria in the system. There are several types of agar, from the richest but not very selective, such as potato-dextrose agar, to some specific for some types of bacteria, such as *Salmonella Shigella* agar, known as SS, which inhibits the growth of most coliforms. On the other hand, there are culture media such as eosin methylene blue agar (EMB agar) that inhibits the growth of Gram-positive bacteria and some non-lactose-fermenting Gram-negative bacteria. Bacteriological cultures were performed for the maximum, minimum, and central points of the concentrations proposed in the experimental design, with duplicates made in each case to ensure data reproducibility in both described agar. The first was nutritive agar, which is characterized by being rich and suitable for the growth of all types of bacteria; the second one, called MacConkey medium, which is selective for the growth of enterobacterium. Bacteriological cultures were developed for the points of greatest

**Table 2 Experiments developed in the factorial design.**

| Experiment | Cd, ppm (A) | Cu, ppm (B) | Hg, ppm (C) | Pb, ppm (D) |
|---|---|---|---|---|
| 1 | 0.10 | 4 | 0.008 | 2 |
| 2 | 0.10 | 4 | 0.008 | 2 |
| 3 | 0.15 | 5 | 0.010 | 3 |
| 4 | 0.05 | 3 | 0.010 | 3 |
| 5 | 0.05 | 3 | 0.006 | 1 |
| 6 | 0.15 | 3 | 0.010 | 1 |
| 7 | 0.15 | 3 | 0.006 | 1 |
| 8 | 0.15 | 5 | 0.010 | 1 |
| 9 | 0.05 | 5 | 0.006 | 3 |
| 10 | 0.15 | 3 | 0.006 | 3 |
| 11 | 0.10 | 4 | 0.008 | 2 |
| 12 | 0.05 | 5 | 0.010 | 3 |
| 13 | 0.05 | 3 | 0.006 | 3 |
| 14 | 0.05 | 3 | 0.010 | 1 |
| 15 | 0.15 | 5 | 0.006 | 1 |
| 16 | 0.15 | 5 | 0.006 | 3 |
| 17 | 0.05 | 5 | 0.010 | 1 |
| 18 | 0.10 | 4 | 0.008 | 2 |
| 19 | 0.10 | 4 | 0.008 | 2 |
| 20 | 0.05 | 5 | 0.006 | 1 |
| 21 | 0.15 | 3 | 0.010 | 3 |

interest to qualitatively complement the proposed factorial design of experiments. Thus, a nutrient agar culture and a selective EMB culture were performed to show the presence of enterobacteria. Additionally, raw water (recently taken from the river), water exposed to the minimum metal concentration in the experimental design (Table 2), and water exposed to the proposed maximum concentration were evaluated.

## Microplastics

The method consists of the filtration of solids obtained through sieves of 5.6 mm and/or 0.3 mm to isolate the solid material of the appropriate size (less than 5 μm). The sieved material is dried to determine the mass of solids in the sample. Solids undergo wet oxidation with hydrogen peroxide (WPO) in the presence of a Fe (II) catalyst to digest organic matter. Plastic waste remains unchanged. The WPO mixture is subjected to density separation at $NaCl_{aq}$ 5M ($d = 1.15$ g/mL) to isolate plastic waste by flotation. Floating solids are separated from denser undigested solids in a density separator. Floating plastic debris is collected in the density separator using a custom 0.3 mm filter, dried at 40 °C, then visually checked under a 40X microscope (*Masura et al., 2015*; *Álvarez-Gómez, 2023*).
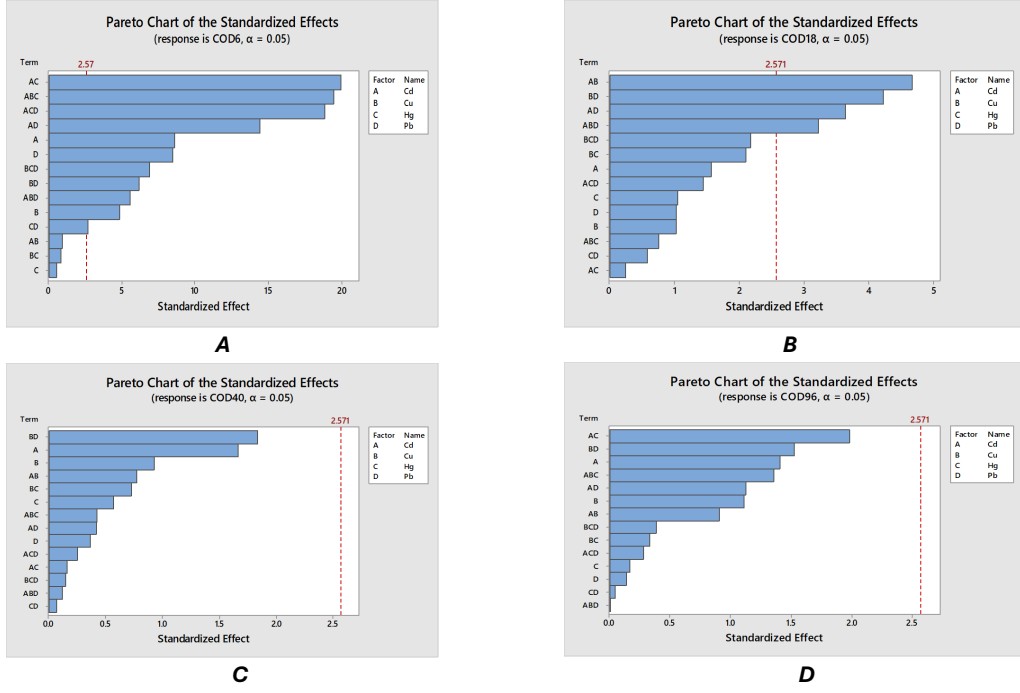

**Figure 2** (A–D) COD at different times.

## RESULTS

### Chemical oxygen demand

According to Pareto diagrams that were made for established measurement times (Figs. 2A–2D), some combinations of metals provoked the greatest impact on the system, such as cadmium-mercury, cadmium-copper-mercury, cadmium-lead-mercury at 6 h. Figure 2A shows that the impact of some combination of metals do not have the same impact and can be concluded to have an inverse effect in respect to time; for longer times the less of impact of to the metals in the system, remaining with an important effect of the combinations of cadmium-copper, copper-lead, cadmium-lead and cadmium-copper-lead. It can be noted that its significance decreases with time to be null at 40 h, which allows proposing that there is adaptation of some microorganisms to the presence of metals. Finally, at 96 h, there is no longer any significance (Fig. 2D).

Pareto charts operate with a significance value, when the interaction of many factors is over this value (the red line) is considered the combination of these factors will impact in the experiment proposed. Thus, *e.g.*, the greatest impact in the COD at 6 h is the interaction cadmium and mercury. Here, factor A corresponds to the cadmium in the system, factor b represents copper, factor c includes all the mercury and factor d represents the lead. In every experiment from the design a different amount of metals was added in the system to evaluate the capacity of the microorganism to adapt. The experiments developed in this study are showed in the Table 2. Furthermore, Pareto diagrams present the effect of the combination of these metals (*e.g.*, Cd-Cu presented as a combination of factors AB).

In Fig. 2A is presented the relationship between the effects of metals on the experimental COD determination. Here, the combination of cadmium and mercury is the most harmful, reaching a value of approximately 20 units on the Pareto chart. This is followed by the combination of cadmium, copper, and mercury, as well as cadmium, mercury, and lead, highlighting the presence of cadmium in each of these combinations. Thus, this metal has the most rapid impact on the microbial development of the system. At 18 h (Fig. 2B), the metal combinations with the greatest impact according to the Pareto chart are cadmium-lead, copper-lead, cadmium-copper, and copper-mercury-lead, although with a lower impact, barely exceeding three units on the Pareto chart. It is important to note that the combination with the greatest impact barely surpasses five units, which demonstrates a drastic reduction in the impact of metals on microorganisms over time.; The comparison between the different Pareto charts shows a significant reduction in the impact of heavy metals on microorganisms after 6 h. Thus, by the time the 18-hour sample is taken, the effect of these metals will be smaller.

Finally, the ratio of metals in the adjustment of COD data at 40 h is presented in Fig. 2C. It is observed that both cadmium and mercury continue to negatively affect the COD data adjustment, however, its significance on the values of this response variable is already nil, while pareto chart value is around 2. The looking of this point is important, since the water contaminated with these metals can be treated by these microorganisms.

### pH profile

A good aerobic operation produces carbon dioxide, as a residue from the degraded organic matter, which when carbonic acid that will tend to acidify the medium. At 0 h metals were added as a soluble compound, nitrates and chlorides; this causes the pH to increase drastically at the beginning. After the second day, the pH begins to decrease.

A Pareto diagram to compare the pH is shown in Fig. 3, which shows that no combination between metals demonstrates the necessary significance for this case study (Pareto chart value > 2.5).

### Sedimentable solids

According to the obtained results of change in the sedimentable solids (SS), it can be observed that after the first day of the contact of the microorganisms with the metal, their growth is little and slow; and even in some experiments (2 and 3) there is destruction of sludge, which means that sedimentation takes longer or even particles are suspended (Table 3).

The effect of metals on the settleable solids was higher on the third day of study than at the beginning of it; these compounds adhere to the floccule at first destroying it although according to the results show favours flocculation in some cases after 48 h. In Fig. 4 the effects of metals on bacteriological growing after 24 h at shown.

### Particle size distribution of floccules in the WWTP

Particle size distribution is an important physical characteristic of wastewater as it develops information about watercourses in sedimentation tanks, chlorine contact tanks and other treatment units. Samples from the WWTP were provided as suspension; particle size

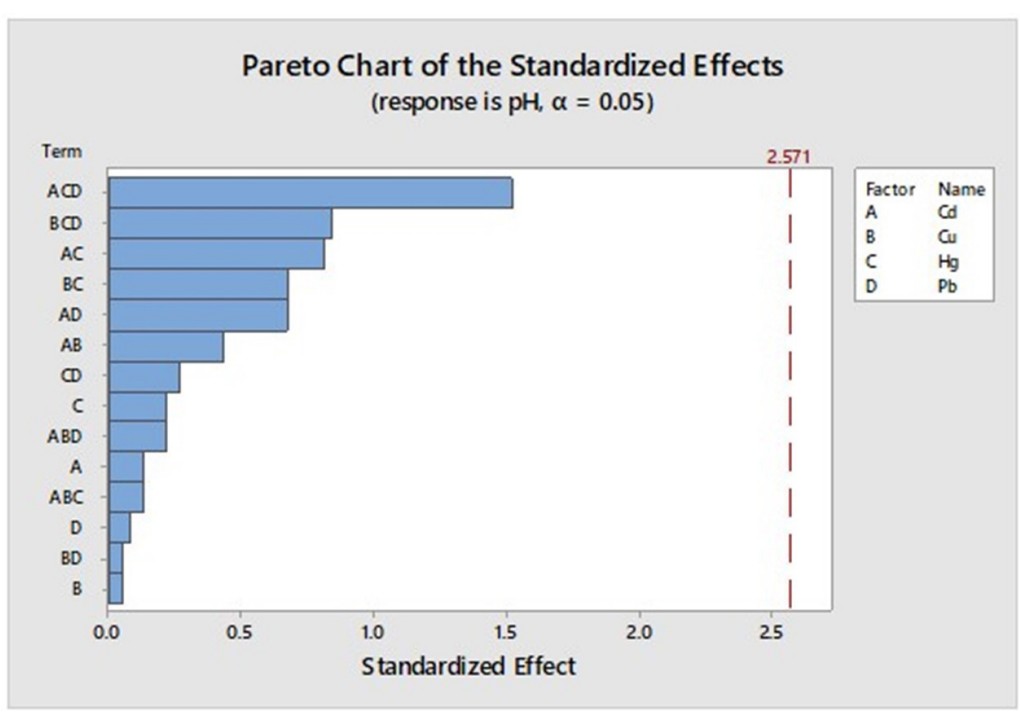

**Figure 3** PH impact in the system.

distribution was determined using the technique described in 'Microbial tolerance to heavy metal contaminants in the sludge' (Table 4, Fig. 5).

## Qualitative bacteriological culture

Once the results of indirect measurement of microbial activity were obtained, the presence of microorganisms and measure of their resistance to metals was qualitatively verified using four specific experiments: raw water, minimum and maximum concentrations of metals and maximum concentrations. In crops from raw water on nutrient agar (Fig. 6A), a large quantity of bacteria was obtained, which are present in domestic wastewater.

## Microplastic presence

The microplastics present in El Silencio River were identified in the area. There are no limits or standards for microplastics in Mexico. Nevertheless, procedures indicated in the NOOA (*Masura et al., 2015*) were followed for the determination of microplastics. The characterization of these microparticles was not carried out; however, it was a qualitative study. The objective of the evaluation of the presence of microplastics is qualitative and exploratory since this reveals the need to strengthen research on the subject in the region, since it is not a topic that still represents relevance for the population that resides there. However, the problem exists and it is better to study it as soon as possible.

**Table 3  Experimental results of sedimentable solids.**

| Experiment | SS @ $t = 0$ Day (ml/L) | SS @ $t = 1$ Day (ml/L) | SS @ $t = 2$ Day (ml/L) | SS @ $t = 3$ Day (ml/L) |
|---|---|---|---|---|
| 1 | 35 | 40 | 60 | 62 |
| 2 | 35 | 40 | 60 | 62 |
| 3 | 35 | 40 | 58 | 62 |
| 4 | 35 | 60 | 58 | 64 |
| 5 | 35 | 52 | 56 | 54 |
| 6 | 35 | 60 | 68 | 60 |
| 7 | 35 | 58 | 72 | 60 |
| 8 | 35 | 60 | 62 | 50 |
| 9 | 35 | 64 | 58 | 58 |
| 10 | 35 | 62 | 58 | 58 |
| 11 | 35 | 46 | 58 | 60 |
| 12 | 35 | 60 | 60 | 70 |
| 13 | 35 | 64 | 60 | 70 |
| 14 | 35 | 68 | 72 | 74 |
| 15 | 35 | 70 | 70 | 80 |
| 16 | 35 | 80 | 72 | 80 |
| 17 | 35 | 60 | 54 | 58 |
| 18 | 35 | 70 | 72 | 70 |
| 19 | 35 | 70 | 72 | 72 |
| 20 | 35 | 62 | 64 | 80 |
| 21 | 35 | 70 | 72 | 82 |

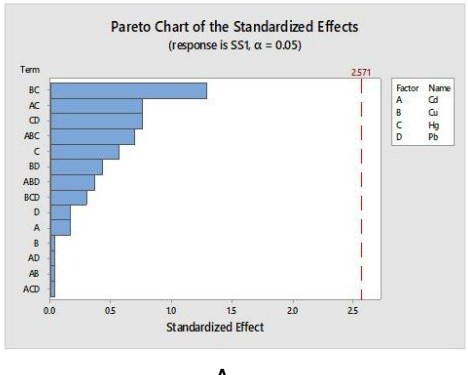

A

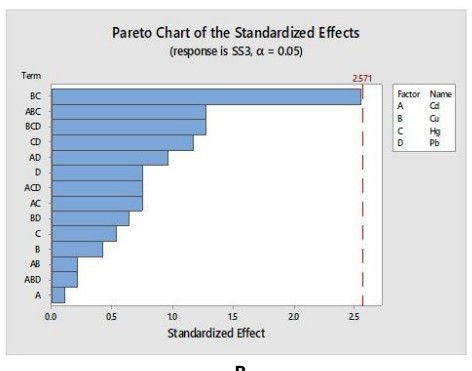

B

**Figure 4  (A–B) Sedimentable solids at different times.**

# DISCUSSION

According to the analysis of the experiments, when the metals come into contact with the system, they quickly affect and cause great impact during a period of time greater than 18 h and less than 40 h; therefore, inhibition or death of microorganisms occur in this period of time. However, as shown in bacteriological cultures, some microorganisms survive even

**Table 4  Particle size distribution.**

| | |
|---|---|
| Average | 23.42 μm |
| Median | 30.23 μm |
| Average/Median | 0.775 |
| standard deviation | 2.540 |
| Variance | 6.449 |
| Bias | −1.781 bias to the left |
| Kurtosis | 3.498 Leptocurtic |

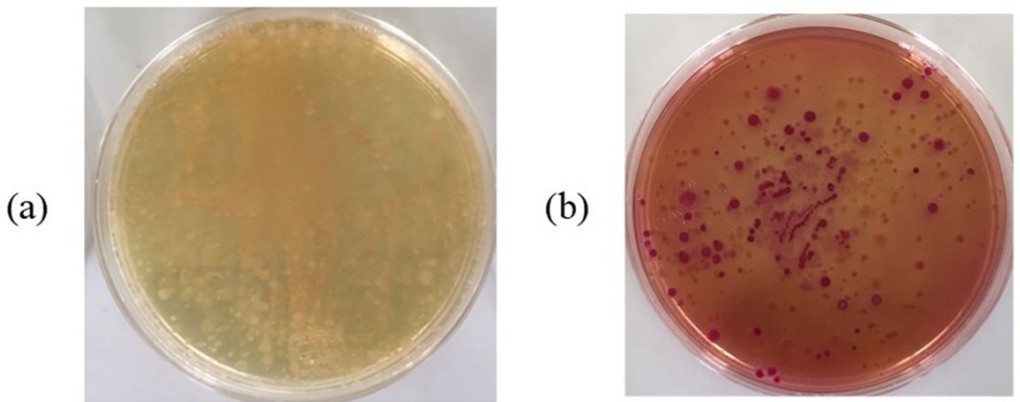

**Figure 5  (A–B) Bacteriological culture of raw water.**

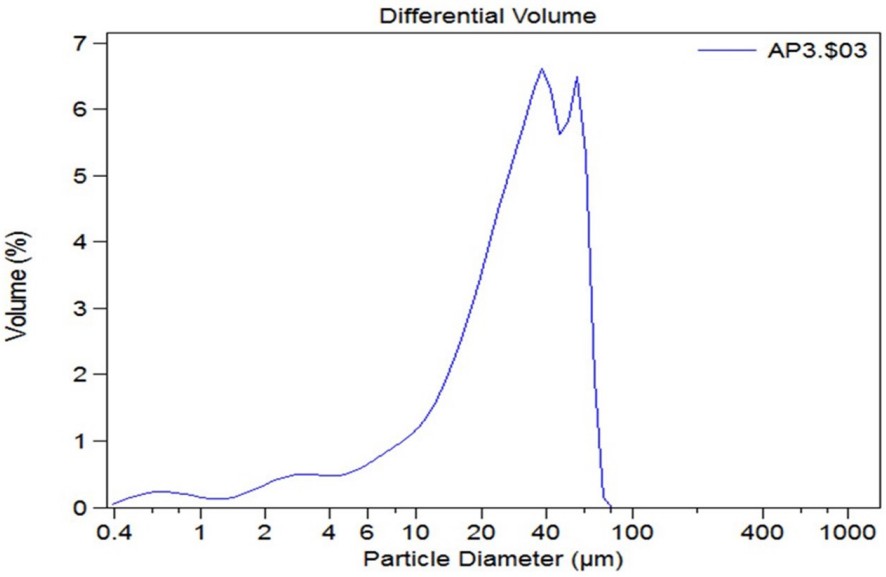

**Figure 6  Particle size distribution.**

at the maximum concentration of metals, so it can be inferred that these microorganisms are adaptable and can continue to be useful for the treatment of water residuals in study. Therefore, even at maximum concentration of metals, microorganisms are able to adapt and adequately reduce the levels of COD, pH and conductivity; this situation guarantees that the water treatment of this system is successful.

It was found that cadmium is the most harmful metal for microorganisms, either alone or in combination with copper and/or lead according to Pareto charts of COD. It was confirmed that copper affects microbial growth, which is known for its bactericidal effect (*Domek, Le-Hevallier & Mcfeters, 1984*). Finally, no significant effect of mercury on the microorganisms was found in the range of concentrations used. The sedimentable solids data obtained reaffirms the information of the decay index LD50 (*Ramalho, 2012*; *Tchobanoglous & Crites, 2000*), which states that in a period not greater than 48 h, the microorganisms are reduced by half.

For the sedimentable solids, a negative effect of the presence of the metals in the system was the inhibition in the growth of flocs, which is noted in the slow increase in this parameter; Mainly seen in experiments with metal concentration higher than the central point (Table 3). However, after the first day of the experiment, the microorganisms are able to adapt and grow slowly. In some experiments the microorganisms adapt quickly, for example, in experiment 20 over 3 days, there was already twice as much sludge. In contrast, experiments 7 and 8 show the slowest growth, while experiments 9 and 10 show good growth on the first day followed by a decrease, and only experiment 9 manages to remain without any change for 48 h. This indicates that the microorganisms are affected in their operation and reproduction, so the quality of treated water may not be the required one. The flocculation of activated sludge indicates the growth of an aerobic process and adequate bacterial growth. When the sludge floats, it is a sign of bacterial destruction and a process lacking the necessary dissolved oxygen. Thus, Figs. 4A and 4B show that metals do not have a significant impact after 24 h of contact with the system. However, the combination of copper and mercury does reach a level of significance on the Pareto chart (value >2.571) after 72 h in the system.

The activated sludge containing this bacterial population would exhibit good functioning in the degradation of organic matter. Analysing the growth in MacConkey agar (Fig. 6B) abundance of enterobacteria was observed; mainly *Escherichia coli* and *Salmonella spp,* which would help the operation of the WWTP, but which are pathogenic to humans. The presence of metals at minimum concentration sensitively affects the growth of bacterial colonies within the selected media; in both media less than half of the colonies are observed in raw water. In addition, the colonies present are smaller, which means that the consumption of nutrients is slower. Subsequently bacteriological cultures were performed at the point of maximum concentration, to identify the remaining microorganisms. Enterobacterium do not adapt to the presence of metals, so they die according to the McConkey agar (Fig. 7B), while in nutritive agar (Fig. 7A) other types of microorganisms still proliferate in colonies of good size.

For the pH measurement it was determined that, perhaps for the short time of experimentation, it could be said that it is not a conclusive parameter when predicting the

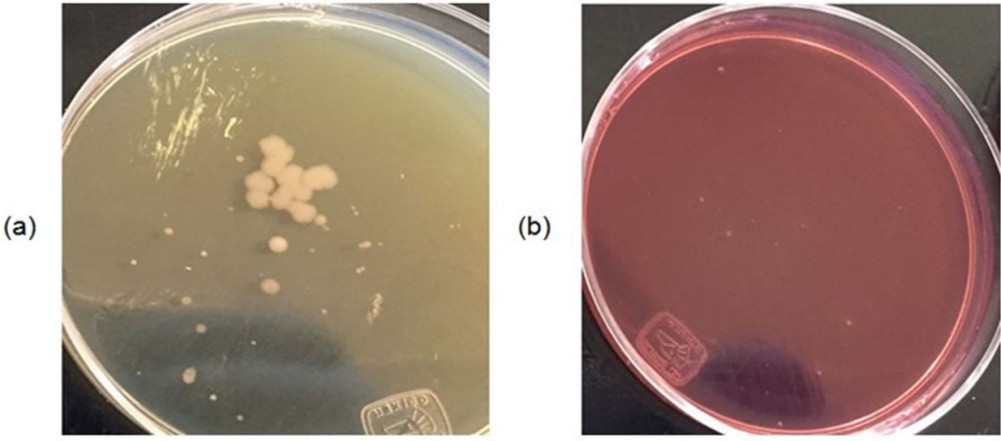

**Figure 7** (A–B) Bacteriological culture at maximum amount of metals.

effect of metals on microorganisms. However, this variable must be followed regularly in the operation of a WWTP.

## CONCLUSIONS

There are combinations of metals that most significantly affect microorganisms, indirectly causing inhibition in the reduction of chemical oxygen demand. In addition, the presence of metals causes a decrease in the amount of sludge, inferred by the sedimentable solids; at least between 24 h and 48 h after the start of the experiment. Subsequently, by adapting to the presence of metals, the remaining microorganisms continue to degrade contaminants, as inferred by the COD decay profile.

According to the analysis of the experiments, when the metals are introduced the system, they quickly affect and cause great impact during a period greater than 18 h and less than 40 h, so that inhibition or death of microorganisms occur in this period. However, as shown in bacteriological cultures, some microorganisms survive even at the maximum concentration of metals. Thus, the microorganisms are adaptable and can still be useful for wastewater treatment.

Microplastics is becoming relevant in the study of water contamination; however, currently there are many uncertainties in how harmful it can be. Further research needs to be developed in this area to make correlation between toxicity effects in public health and the complicity of water treatment. There are possible interactions with microplastics and trace elements, and this combination could complicate the function of the activated sludges in the WWTP.

## ACKNOWLEDGEMENTS

We greatly appreciate the facilities granted by the staff of the WWTP of the Salvador Escalante municipality in obtaining water to carry out this study. Analytical analysis from

laboratories of wastewater and environment at the Universidad Michoacana de San Nicolás de Hidalgo are greatly acknowledged.

### Funding
This work was supported by the Consejo Nacional de Humanidades Ciencia y Tecnología (CONAHCYT), CVU 861765, CVU 861753. The funders had no role in study design, data collection and analysis, decision to publish, or preparation of the manuscript.

### Grant Disclosures
The following grant information was disclosed by the authors:
the Consejo Nacional de Humanidades Ciencia y Tecnología (CONAHCYT): CVU 861765, CVU 861753.

### Competing Interests
The authors declare there are no competing interests.

### Author Contributions
- Mario Alberto Pérez-Méndez conceived and designed the experiments, performed the experiments, analyzed the data, prepared figures and/or tables, and approved the final draft.
- Guadalupe Selene Fraga-Cruz conceived and designed the experiments, performed the experiments, prepared figures and/or tables, and approved the final draft.
- Gustavo Álvarez-Gómez performed the experiments, prepared figures and/or tables, and approved the final draft.
- Fabricio Nápoles-Rivera analyzed the data, authored or reviewed drafts of the article, and approved the final draft.
- Gladys Jiménez-García analyzed the data, authored or reviewed drafts of the article, and approved the final draft.
- Rafael Maya-Yescas performed the experiments, analyzed the data, authored or reviewed drafts of the article, and approved the final draft.

### Data Availability
The raw data have been uploaded to the Supplemental Files.

### Supplemental Information
Supplemental information for this article can be found online at http://dx.doi.org/10.7717/peerj.18531#supplemental-information.

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
