# Peer review of "Analysis of the anthropogenic effect on the Silencio River in Salvador Escalante, Michoacán, México"

_PeerJ, doi:10.7717/peerj.18531_

## Round 0.1 · original submission · Major Revisions

The authors need to revise the manuscript meticulously. Both the reviewers have provided very useful comments which will help the authors to revise the manuscript.

Reviewer 1 ·

Basic reporting

I strongly suggest to do a better literature review. I have made suggestions throughout the text in the attached pdf. Please do the necessary corrections. Additionally, please look at how the information is written in other papers

Search WWTP microbes in scholar.google.com to see how such papers can be written.

Experimental design

You need to write in details as to what you did. Please remove the explanations and write only what you did.

Validity of the findings

Write only what you have found in your experiment.

Additional comments

Add all other explanations in the discussion section. Discussion can not be one paragraph

Annotated reviews are not available for download in order to protect the identity of reviewers who chose to remain anonymous.

Reviewer 2 ·

Basic reporting

Pérez-Méndez analysed anthropogenic effect on Silencio river in Mexico. I find the study interesting with detailed background on the impact of wastewater treatment plants on microogranisms and health of environment.
The manuscript provided many details on the heavy metals and discharge quality of the effluents. The article structure is informative. The abstract needs to be in line with the main question analysed in the study.

Experimental design

The 24-1 factorial design of the study needs more details. There are not much details provided on qualitative methods followed to collect data on heavy metals, oxygen content etc.
The data analysis needs more details.

Validity of the findings

No comments

Additional comments

It is not clear how useful these finding are considering there are previous studies conducted on this system.

---

## Round 0.2 · Major Revisions

I would suggest the authors to revise the manuscript carefully. The results section need lot of restructuring and addition of graphs/representation. The data needs to be integrated with the findings.

Reviewer 1 ·

Basic reporting

The introduction still lacks a connection between the paragraphs. Each paragraph currently talks about a discrete topic. There has to be connecting lines between the paragraphs.

What is the relevance of section 2.2 in the methods sections? This kind of information should go into the discussion or introduction sections.

There is a mix of active and passive voice throughout the text. Please fix. Sometimes hour is written as h and other times as hour. Please stick to one format.

3.3 lacks sufficient information

3.4 The results do not contain data. If a reader reads the paragraph without seeing the graphs, they will not get any idea of what is happening. The measured data (numbers) need to be written out in the paragraphs.

Experimental design

The manuscript currently lacks any experimental design. The authors have shown culture plates but the methods provide no information on it. The site map has not been cited in the text.

Validity of the findings

It is difficult to comment based on information currently provided.

Annotated reviews are not available for download in order to protect the identity of reviewers who chose to remain anonymous.

---

## Round 0.3 · accepted · Accept

The paper has been satisfactory revised revised and can now be accepted.